# Histomorphometric Analysis of 38 Giant Cell Tumors of Bone after Recurrence as Compared to Changes Following Denosumab Treatment

**DOI:** 10.3390/cancers15174249

**Published:** 2023-08-24

**Authors:** Sophia Arndt, Wolfgang Hartmann, András Rókusz, Benedikt Leinauer, Alexandra von Baer, Markus Schultheiss, Jessica Pablik, Hagen Fritzsche, Carolin Mogler, Imre Antal, Daniel Baumhoer, Kevin Mellert, Peter Möller, Miklós Szendrői, Gernot Jundt, Thomas F. E. Barth

**Affiliations:** 1Institute of Pathology, University Ulm, 89081 Ulm, Germany; 2Gerhard-Domagk-Institute of Pathology, University Hospital Münster, 48149 Münster, Germany; 3Institute of Pathology and Experimental Cancer Research, Semmelweis University, 1085 Budapest, Hungary; 4Clinic for Trauma, Hand, Plastic and Reconstructive Surgery, University Hospital Ulm, 89081 Ulm, Germany; 5Institute of Pathology, University Hospital Carl Gustav Carus, 01307 Dresden, Germany; 6Centre for Orthopedics, Trauma and Plastic Surgery, University Hospital Carl Gustav Carus, 01307 Dresden, Germany; 7Institute of Pathology, Technical University of Munich, 81675 Munich, Germany; 8Institute of Orthopedics, Semmelweis University, 1085 Budapest, Hungary; 9Bone Tumor Reference Centre at the Institute of Pathology, University Hospital Basel and University of Basel, 4003 Basel, Switzerland

**Keywords:** giant cell tumor, recurrence, denosumab, malignant transformation, immunohistochemical profiling

## Abstract

**Simple Summary:**

Giant cell tumor of bone (GCTB) is an osteolytic tumor driven by an H3F3A-mutated mononuclear stromal cell with an accumulation of osteoclastic giant cells. Anti-RANKL antibody therapy with denosumab leads to morphological changes, and sarcomas have been described in association with therapy. We compared tissue from patients with recurrence of GCTB with samples after denosumab therapy, including two cases of malignant transformation. We detected that profound changes in morphology and the immunohistochemical profile after denosumab therapy are not compatible with changes detected in the sarcomas including expression of RUNX2, SATB2, and KI-67.

**Abstract:**

Giant cell tumor of bone (GCTB) is an osteolytic tumor driven by an *H3F3A*-mutated mononuclear cell with the accumulation of osteoclastic giant cells. We analyzed tissue from 13 patients with recurrence and 25 patients with denosumab therapy, including two cases of malignant transformation. We found a decrease in the total number of cells (*p* = 0.03), but not in the individual cell populations when comparing primary and recurrence. The patients treated with denosumab showed induction of osteoid formation increasing during therapy. The total number of cells was reduced (*p* < 0.0001) and the number of *H3F3A*-mutated tumor cells decreased (*p* = 0.0001), while the *H3F3A* wild-type population remained stable. The KI-67 proliferation rate dropped from 10% to 1% and Runx2- and SATB2-positive cells were reduced. The two cases of malignant transformation revealed a loss of the *H3F3A*-mutated cells, while the KI-67 rate increased. Changes in RUNX2 and SATB2 expression were higher in one sarcoma, while in the other RUNX2 was decreased and SATB2-positive cells were completely lost. We conclude that denosumab has a strong impact on the morphology of GCTB. KI-67, RUNX2 and SATB2 expression differed depending on the benign or malignant course of the tumor under denosumab therapy.

## 1. Introduction

Giant cell tumor of bone (GCTB) is an osteolytic tumor accounting for about 4–5% of all primary bone tumors with a peak incidence between 20 and 45 years [1]. GCTB usually occurs in the metaphyseal/epiphyseal region of the long tubular bones of the mature skeleton. The bones most affected are the distal femur and proximal tibia. Further frequently reported localizations are the proximal humerus and the distal radius. In the axial skeleton, GCTB occurs most commonly in the sacrum and in the vertebral body [1,2]. GCTB is classified by the WHO as a primary bone tumor with locally aggressive growth (ICD-O code 9250/1). Between 15% and 50% of GCTB recur locally after curettage usually within two years. Dissemination is rare (2–5%) and almost exclusively reported in the lungs [3,4]. Primary malignancy in a giant cell tumor has been described by the WHO as a high-grade sarcoma arising in an otherwise conventional GCTB. Malignant transformation in a GCTB is more common after treatment of a conventional GCTB, including radiotherapy and the conventional GCTB may or may not be detectable in these cases [1].

There are at least three cellular components in GCTB that are important for understanding its biology. First, the neoplastic tumor cell of GCTB which is mononuclear and spindle-shaped. In addition, there is a population of macrophage-like, mononuclear and non-neoplastic stromal cells and osteoclastic giant cells. The giant cells are derived from hematopoietic stem cells that develop into monocytes and are then attracted to the tumor environment by macrophage colony-stimulating factor (M-CSF) secreted by the GCTB mononuclear cell compartment. Osteoclast progenitor cells express RANK (Receptor Activator of NF-κB), and the interaction of RANK and its ligand RANKL (Receptor Activator of NF-κB Ligand) leads to osteoclast differentiation. In GCTB, abundant RANKL expressed and secreted by neoplastic stromal cells leads to the induction of giant cell differentiation [5,6].

In 2013 a histone *H3F3A* mutation was identified as a highly specific driver mutation for GCTB. *H3F3A* encodes for the histone variant H3.3 and the mutation involves Glycine 34 in 90%, with amino acid exchange G34W (p.Gly34Trp). This finding is crucial in diagnostics, as an antibody was developed that binds specifically to the mutated p.G34W side of histone H3.3 in the nuclei of the neoplastic cells [7,8,9,10]. Furthermore, the *H3F3A* mutation and the resulting alteration in H3.3 nucleosome biology leads to epigenetic changes such as chromatin methylation and telomeric instability. These findings have been shown to be involved in uncontrolled RANKL transcription activity [6].

With regard to therapy, surgical removal is the gold standard. Either curettage with subsequent covering of the defect by polymethylmethacrylate (PMMA) or resection with joint replacement are performed. Systemic therapy is considered in cases where tumors are large or localized in topographic regions like the pelvis to improve the surgical outcome. Possible drug treatments include the humanized anti-RANKL antibody denosumab, bisphosphonates, and interferon-α [2,11].

Anti-RANKL therapy is based on the inhibition of the interaction of RANKL and the RANK receptor. Denosumab is a decoy receptor of RANKL and thus has a functional effect analogous to the physiological inhibitor osteoprotegerin. GCTBs treated with denosumab show significant morphological changes with a reduction in giant cells leading to reduced osteolysis. The described morphological changes are an increase in the formation of sclerosed bone tissue, whereby *H3F3A*-mutated tumor cells can persist in the sclerosed bone [11,12]. In contrast, bisphosphonates as an alternative treatment regimen in GCTB act by direct inhibition of osteoclastic cells. Although known primarily for the treatment of osteoporosis, bisphosphonates have been shown to result in increased bone mineralization, increased apoptotic index, and decreased giant cell numbers in GCTB [11,13].

For therapy, denosumab 120 mg is administered subcutaneously every week in the first three weeks and is then applied at a dose of 120 mg every four weeks [14]. Currently, there is no consistent recommendation as to how long denosumab should be administered [15,16]. Furthermore, there have been reports of malignant transformation of GCTB associated with denosumab therapy; this led to a manufacturer’s information letter in 2018 reporting on a large-scale double-blind study regarding the association of malignancies with denosumab compared to therapy with zoledronic acid, revealing a mild increase in malignancy in GCTB after denosumab therapy (1.5% versus 0.7%) [17,18]. Regarding the mechanisms of malignant transformation in GCTB it has been shown that all malignant tumors carried at least one extra driving mutation and displayed genetic characteristics similar to osteosarcomas, such as a high number of mutations, added driver events, and significant aneuploidy. Through analysis of differential methylation profiling, CCND1, responsible for encoding cyclin D1, emerged as a potential cancer-driving gene in these tumors due to the distinctive hypermethylation of the CCND1 promoter observed specifically in GCTBs [19].

## 2. Materials and Methods

### 2.1. Patients

We collected paraffin tissue blocks from 38 patients with giant cell tumor of bone. All tumors tested positive for the *H3F3A* mutation in immunohistochemical analysis. Patients were included who either had a recurrence (*n* = 13) after initial curettage or who were treated with denosumab (*n* = 25). All patients gave their written consent. The study was approved by the Ethics Committee of the University of Ulm (authorization for the use of archived human material 03/2014) and is in line with the Declaration of Helsinki.

### 2.2. Histology and Immunohistochemistry

The formalin-fixed tissue samples were mounted on slides in 3–4-µm-thick sections. Hematoxylin and eosin staining was carried out using the standard protocol. We performed immunohistochemistry using an avidin-biotin complex method and the K005 AP/RED Detection System (cat# K5005; Dako, Santa Clara, CA, USA). As primary antibodies we used Anti-Histone H3.3 G34W Rabbit Monoclonal Antibody, Clone RM263 (cat# 31-1145-00; RevMAb Biosciences, San Francisco, CA, USA; 1:800); KI-67 Antigen, Clone MIB-1 (cat# M7240, Dako, Hamburg, Germany; 1:200) and Anti-Runx2 (F-2) Antibody (cat# sc-390351; Santa Cruz Biotechnology, Dallas, TX, USA; 1:50). Anti-SATB2 Clone EP281 (cat#384R-10, Cell Marque, Rocklin, CA, USA; 1:25) was stained with DAKO Autostainer. For KI-67, SATB2 and RUNX2 it was not possible to perform immunohistochemical staining for all cases due to lack of tissue. (KI-67 primary before recurrence: *n* = 12; KI-67 recurrence: *n* = 12; SATB2 pre-denosumab: *n* = 22; SATB2 post-denosumab: *n* = 21; SATB2 primary before recurrence: *n* = 11; SATB2 recurrence: *n* = 10; RUNX2 pre-denosumab: *n* = 21; RUNX2 post-denosumab: *n* = 19; RUNX2 primary before recurrence: *n* = 11; RUNX2 recurrence: *n* = 10). Details can be seen in Appendix A.

To achieve the highest accuracy in quantifying the total number of *H3F3A*-mutated cell nuclei and to distinguish reliably between *H3F3A* wild-type stromal cell nuclei and *H3F3A* wild-type giant cell nuclei, we used the open-source software QuPath version 0.3.0 [20] and counted the cell nuclei in three representative microscopic fields (Appendix A). For further immunohistochemical characterization of the stromal cells we switched to semi-quantitative analysis and evaluated the percentage of KI-67-, SATB2-, and RUNX2-positive cells using a multi-headed microscope. All samples were analyzed by two experienced pathologists in a blinded fashion. Beforehand, we compared the computer-based method by QuPath with our semi-quantitative evaluation using a multiheaded microscope and came up with a difference of less than 5% in all cases.

### 2.3. Statistics and Graphs

The *p*-value was calculated using the two-sided paired Student’s *t*-test. The diagrams and calculations were created with Excel and Prism Graphpad.

## 3. Results

### 3.1. Clinic

We established a group of 13 denosumab-naive patients with corresponding recurrences and a group of 25 patients with samples before and after denosumab therapy, including two patients whose samples showed malignant transformation.

The studied group with recurrences consisted of 7 males and 6 females (average age 34.6); the group with denosumab therapy included 14 males and 11 females (average age 33.6). The localization of the primary tumors was distributed in the group with recurrences as follows: tibia: *n* = 6, femur: *n* = 2, radius: *n* = 2, os metacarpale IV: *n* = 1, left hand, phalanx, digitus 1: *n* = 1 humerus: *n* = 1. Regarding the group of patients with denosumab treatment, the tumors were localized in the tibia: *n* = 8, femur: *n* = 9, radius: *n* = 3, left hand, phalanx, digitus 3: *n* = 1, hip: *n* = 2, patella: *n* = 1, and thoracic vertebral body: *n* = 1.

The duration of denosumab therapy ranged from a single dose to 25 months. Details regarding denosumab therapy and clinical data are summarized in Table 1.

### 3.2. Histological Findings

The samples of all primary tumors in this study showed a typical morphology of GCTB, with osteoclast-like giant cells with up to 40 nuclei intermingled with a mononuclear cell population with round- to oval-shaped nuclei without atypia. Regarding recurrences, we detected no change in the number of giant cells and a decrease of 16.5% of the total number of cells during the time course that was significant (*p* = 0.03).

After treatment with denosumab, the morphology of GCTB changed dramatically. All cases showed an almost complete loss of giant cells (Figure 1b; Appendix A). We next counted the nuclei of the mononuclear cells before and after denosumab treatment; the cellular density was considerably reduced, with the reduction in the total number of cells averaging 40.1% (*p* < 0.0001) (Figure 1a).

As a further morphological feature after denosumab treatment, we observed areas of matrix neoformation and distinguished between two types. The first type showed a strand-like matrix formation with intermingled spindle-shaped cells and can be further subdivided into two patterns: areas with a high cell density of spindle cells without atypia or scarring fibrosis with a drastic decrease in cells. The second main type of matrix formation detected revealed bone neoformation that was either osteoid, showing osteoid trabeculae with rimming osteoblasts, or roughly organized osteoid (Figure 2). If both main types were detected simultaneously in a tissue probe; we evaluated the percentage of the types of matrix formation and assigned each sample to the group that accounted for more than 50%. Strand-like matrix formation with spindle-shaped cells was the predominant type in 10 tissue samples. Osteoid formation was the predominant type in 11 patients. Two patients could not be assigned to either group because they expressed both types equally. We matched the patterns with the duration of denosumab therapy that each of the patients had received, but could not find a correlation. The two tumors that developed sarcoma under denosumab therapy were excluded from this analysis and will be discussed in detail in the next section.

We analyzed two cases who had developed a sarcoma during denosumab treatment. The tumor presentation of patient 24 has been described in detail in previous publications [18,21]. In brief, the primary tumor showed a typical morphology of GCTBs with osteoclast-like giant cells with intermingled mononuclear stromal cells. After treatment, a high-grade pleomorphic sarcoma was diagnosed showing necrosis and high mitotic activity. The histology of patient 25 was as follows: the primary tumor had a typical morphology of GCTB with giant cells and round-shaped stromal cells without atypia. However, we noticed areas with an increase in spindle cells already present before denosumab treatment, showing mononuclear cells with a distinct nucleolus and a decrease in giant cells. The osteosarcoma after denosumab therapy revealed a complete loss of giant cells. The morphology revealed sheets of blastic cells with prominent nuclei, mitotic figures including atypical mitotic figures with up to five mitoses per high power field and lace-like malignant osteoid with infiltration of tumor cells into the surrounding areas, as well as areas with focal chondroblastic differentiation (Figure 3).

### 3.3. Immunohistochemical Findings

#### 3.3.1. H3F3A G34W Staining

In the samples from patients who did not receive denosumab, we compared cell populations of the primary tumor with those of the first recurrence. For this purpose, we stained the tissue with the mAB specific for H3.3 G34W and counted the nuclei of *H3F3A* G34W-stained cells, the *H3F3A* G34W-negative mononuclear cells, and giant cells. When comparing primary tumor and recurrence, a decrease in the total number of cells by 16.5% was observed (*p* = 0.03). This slight decrease was not significant in the *H3F3A*-mutated or wild-type cell population, which means that no tendency to increase or decrease was detectable in the mononuclear cell compartment (Appendix A).

Next, we compared pre- and post-denosumab samples of the 23 patients treated with denosumab (Figure 4a; Appendix A). Almost all samples showed a complete loss of giant cells. Further, we detected a drop of 31.5% in the total number of mononuclear stromal cells (*p* = 0.0009), showing that not only the reduction in giant cells is responsible for the decrease in the total number of cell nuclei (Figure 5a). Analysis of the individual cell populations revealed that the *H3F3A* G34W-positive cell nuclei were significantly reduced by 41.8% (*p* = 0.0001), while the number of *H3F3A* G34W-negative cell nuclei remained stable (Figure 5b,c). This results in a shifted ratio of the cell populations: before denosumab treatment, the mutated cell nuclei had a ratio of 60.9% to 39.1% of wild-type cell nuclei. After treatment, the ratio was 48.7% to 51.3% (Figure 5d). One single sample (patient 2) revealed a complete loss of *H3F3A G34W*-positive cell nuclei after being treated with denosumab for 25 months.

The two cases of malignant transformation during denosumab treatment (patient 24 and patient 25) revealed a complete loss of giant cells and *H3F3A* G34W-positive cells (Appendix A).

#### 3.3.2. Proliferation Rate

Comparing the KI-67 staining of the primary and recurrences without denosumab treatment, we detected values ranging from 1% to 20% (mean 5.9%); no significant difference between the primary and the recurrence was detected (Figure 6a; Appendix A). Regarding the distribution of KI-67-positive cells, we noticed a higher KI-67 score of mononuclear cells associated with the foci of giant cells (Figure 4b). The number of these foci did not change in the primary tumors and the recurrences.

When analyzing KI-67 staining before and after denosumab treatment, we found that denosumab treatment was associated with a reduction in the proliferation rate of mononuclear cells (*p* < 0.0001). Samples 1, 6, and 18 were exceptions, with a very low proliferation rate of 1% and less in the primary tumor that remained stable after treatment (Figure 6b; Appendix A).

In contrast, the two patient samples associated with malignant transformation during denosumab treatment showed an increase in KI-67 from 5% to 50%, and from 5 to 60% in the malignantly transformed sarcoma after denosumab therapy (Appendix A).

#### 3.3.3. SATB2

Regarding expression of Special AT-rich sequence-binding protein 2 (SATB2), all samples of GCTB showed a high nuclear expression of SATB2 limited to the mononuclear component, while giant cells were SATB2-negative in all samples analyzed. However, the percentages of SATB2-positive mononuclear cells differed largely in the primary tumors, ranging from 10% to 80%. In samples with recurrences, the number of SATB2-positive cells ranged from 10% to 60% (Figure 6c; Appendix A).

In contrast, in denosumab-treated samples, we noticed a decrease in the proportion of SATB2-positive cells in all 23 samples from about 60% of SATB2-positive cells before denosumab treatment to 20% after denosumab treatment (*p* < 0.0001; Figure 4c and Figure 6d; Appendix A). We analyzed these changes in correlation to the duration of denosumab therapy and failed to detect any correlation between SATB2 expression and the duration of denosumab therapy.

Analysis of a malignant transformation after denosumab therapy showed a complete loss of SATB2-positive cells in the post-denosumab-associated sarcoma of patient 24. In contrast, we found an increase from 50% in the primary to 80% in the sarcoma of patient 25 (Appendix A).

#### 3.3.4. RUNX2

RUNX2 was detected in the nuclei of the stromal component of all 38 samples of GCTB ranging from 10% to 95%; giant cells were consistently negative for RUNX2; RUNX2 expression was detected with a value >/= 90% positive cells in 24 samples of primary tumors; eight samples of GCTB showed a RUNX2 expression below 90% with one sample revealing 10% RUNX2-positive cells. Compared to the group of recurrent GCTB, there was no significant change in the distribution of RUNX2 expression (Figure 6e; Appendix A).

In the group of samples from patients treated with denosumab, we observed a significant decrease in RUNX2 expression in all samples, falling from 90% to 50% on average (*p* = 0.0003) (Figure 4d and Figure 6f; Appendix A).

We did not detect any correlation when analyzing these data for correlation with the duration of denosumab therapy.

For the analysis of a malignant transformation after denosumab therapy in patient 24, only the post-denosumab sample was available. This sarcoma showed a low expression of RUNX2 in 10% of the nuclei of the stromal cells. In the sample of patient 25, RUNX2-positive cells increased from 60% in the primary to 90% in the sarcoma (Appendix A).

## 4. Discussion

GCTB is a complex tumor consisting of various cell populations with osteoclastic giant cells and a mononuclear cell population including *H3F3A*-mutated neoplastic cells. Our aim was to analyze the morphology of the different compartments of GCTB and its recurrence over time for comparison with GCTB treated with denosumab. Therefore, we included samples of patients with primary and recurrent GCTB and GCTB before and after denosumab therapy.

We compared these two groups of GCTB sample pairs: We could not detect any clear morphological differences between primary and its recurrence, but our computer-based cell count showed a slight decrease in the total number of cells. When we analyzed the cell populations individually, however, there was no significant decrease, i.e., they remained stable. Thus, the composition and percentages of the cell populations do not change in the recurrence.

In contrast to this, denosumab leads to dramatic histomorphological changes. As described after denosumab therapy in GCTB, we saw a great reduction in giant cells, a decrease in mononuclear cell density, and an induction of matrix formation [12,22,23,24,25,26]. Although matrix formation was increased in all samples analyzed, the patterns of matrix formation were heterogeneous, ranging from osteoid-like bone formation to scarring fibrosis. Erdogan et al. (2016) described several cases with different histological findings including fibrous tissue and woven bone. They stated that pathologists should be aware of these different patterns to avoid misdiagnosis. They also raised the question of whether there is a correlation between these histopathologic characteristics and the dosage of treatment [25]. In this regard, we were able to categorize histological patterns into two types with two subtypes each in our cohort; however, when we matched these different patterns with the duration of denosumab therapy we could not find any correlation. This may point to an extremely individual and so far unpredictable response to denosumab regarding matrix formation. These findings are supported by a nine-case study conducted by Roitman et al. (2017) which also outlined the extensive diversity in bone formation, encompassing trabeculae and irregular strands of osteoid. These authors also noted a loss of giant cells in most cases but showed four cases in which giant cells were preserved in denosumab-treated samples. The authors conclude that there is no clear correlation between treatment duration and the extent of histologic changes [27]. More recent studies and reviews such as Kumar et al. (2023) and Rheki et al. (2023) showed similar results [28,29].

Studies that analyzed the effect of denosumab therapy on the *H3F3A*-mutated cells show that these cells still persisted in the post-treatment samples [22,23,24,26,30]. We noticed a significant reduction in *H3F3A*-mutated cells after denosumab therapy, including one sample with a complete loss of the neoplastic cells after 25 months of denosumab treatment. Girolami et al. (2016) conducted a study with 15 patients treated with denosumab and described persisting mutated cells; the authors concluded that denosumab had no direct effect on the neoplastic cells, but on the tumor microenvironment [26]. Treffel et al. (2020) evaluated the percentage of *H3F3A*-mutated cells before and after denosumab therapy and found no significant change in the number of the neoplastic cell population after denosumab therapy [24]. Ud Din et al. (2020), in turn, showed a significant decrease in neoplastic cells from 68.8% to 26.9% after denosumab therapy [22]. Our quantification of the total number and percentage of *H3F3A*-mutant cells revealed a decrease of 41.8% of mutant cells confirming these data. Furthermore, we found no change in the number of wild-type cells after denosumab therapy. This indicates that the inhibition of RANKL by denosumab not only inhibits the induction of giant cells but also has an influence on the RANKL-expressing neoplastic cells. This critical influence of denosumab on the cell populations is further supported by the fact that we did not observe any changes in the number of *H3F3A*-mutated cells in the group of patients not treated with denosumab.

We further analyzed the stromal cell compartment in detail and noticed a significant increase in matrix formation after denosumab therapy. Therefore, we analyzed SATB2 and RUNX2 expression as markers of osteoblast differentiation [31,32]. Yang et al. (2022) analyzed SATB2 in GCTB after denosumab treatment and did not see any changes in the SATB2-positive population [23]. Girolami et al. (2016) described a reduction in SATB2-positive cells and no change in the RUNX2-positive cell compartment [26]. Ud Din et al. (2020) observed reduced SATB2-positive cells after treatment, which was not statistically significant. Our study confirmed these observations concerning SATB2 and showed a decrease from 60% to 20%. In contrast to Yang et al. (2022), we noticed a significant reduction in RUNX2 staining in the mononuclear compartment from 90% to 50%.

Since the patients in our analysis were all treated with denosumab for different time periods, we had the opportunity to analyze time-dependent changes during denosumab therapy. Our analysis revealed a high increase in matrix formation including osteoid formation as well as the induction of fibrosis. We detected a decrease in *H3F3A* G34W-positive cells accompanied by a decrease in SATB2-, RUNX2-, and KI-67-positive cells—none of these changes correlated in time with the duration of denosumab therapy. The decrease in SATB2-positive cells as a marker of late osteoblast differentiation is of interest and confirms other studies [22,26]; however, these authors concluded that the SATB2-positive cell pool increases after an initial period of decrease as an initial effect of denosumab therapy. In these studies, patients were treated with denosumab for a median of 5.7 months (Girolami et al. 2016) and 1.5 months (Ud Din et al. 2020) [22,26]. In our study with a median treatment of 8.4 months, this hypothesis cannot be confirmed in correlation analyses and rather suggests that the extent of changes over time is extremely heterogeneous. In our cohort SATB2-positive cells decreased after denosumab therapy; SATB2-positive cells were associated with bone formation and mineralization [22]. Nevertheless, we observed a strong induction of matrix formation accompanied by a change in the immunohistochemical profile of mature osteoblasts: Thus, we conclude that this indicates a possible defect in the maturation of these cells.

The two GCTBs in our study that underwent malignant transformation associated with denosumab treatment both revealed a complete loss of the *H3F3A*-mutated mononuclear cells in the established osteosarcomas. Other reports confirm a similar loss of the *H3F3A*-mutated cell pool. However, other studies have shown the persistence of *H3F3A*-positive cells in the sarcomas associated with denosumab treatment [33]. One explanation for the loss of *H3F3A*-mutated stromal cells during malignant transformation is that the sarcoma arises from a subclone belonging to the *H3F3A* wild-type cell population in the tumor. For example, Hasenfratz et al. (2021) presented a sarcoma arising in a GCTB in which the *H3F3A* mutation was lost after denosumab treatment; however, their molecular analysis of samples before and after denosumab treatment revealed an overlapping mutation for *FGFR1* [18]. This hypothesis is supported by the fact that our study shows the suppressive effect of denosumab on the *H3F3A*-mutated cell population compared with the unaffected wild-type cell population, thereby dramatically altering the tumor environment.

SATB2 and RUNX2 expression differed in the two samples associated with malignant transformation after denosumab therapy. In one sample SATB2 staining was completely lost after denosumab therapy and RUNX2 was at a very low expression level, in contrast to the results of the denosumab-treated group with no malignant transformation. However, in the second sarcoma associated with denosumab treatment, we noticed an increase in these markers after denosumab compared to the sample before treatment. Both sarcomas had in common a high increase in KI-67 staining, pointing to diverging pathways associated with malignant transformation in these samples. The expression of these three markers distinguishes the denosumab-treated tumors with a benign course from those that undergo malignant transformation. However, as we have only analyzed two malignant tumors in our cohort and these two are even inconsistent with each other, this is only an observation so far. This is a limitation of our study and should be further investigated.

## 5. Conclusions

In conclusion, we have shown that changes in the cellular and extracellular compartments induced by denosumab treatment in GCTB are complex and may point to an individual background. We have shown that although the neoplastic population after treatment is reduced, it does persist and is lost only after 25 months of treatment with denosumab. We have shown that the markers KI-67, SATB2, and RUNX2 are generally reduced during denosumab therapy. Changes regarding these patterns, such as an increase in KI-67 after denosumab therapy, may point to a malignant transformation associated with treatment and should indicate caution regarding surveillance of these patients.

## Figures and Tables

**Figure 1 cancers-15-04249-f001:**
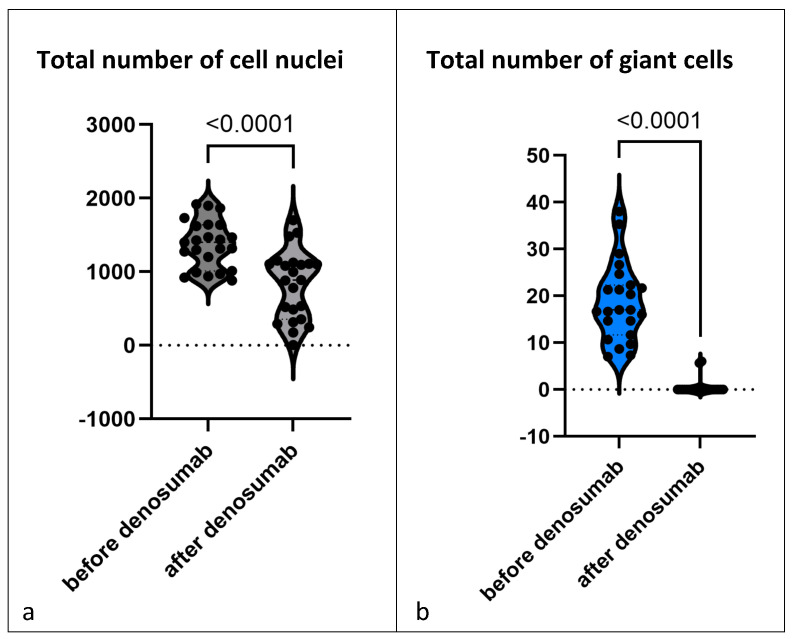
Every individual data point shows the mean of (**a**) total number of cell nuclei and (**b**) total number of giant cells counted in three representative microscopic fields (200× magnification). The graph shows all 23 GCTBs treated with denosumab (but without malignant transformation) and compares before and after denosumab therapy. Violin diagram is used to show the distribution of the data, the individual data points are each the sample of one patient, small dotted lines show median and quartiles.

**Figure 2 cancers-15-04249-f002:**
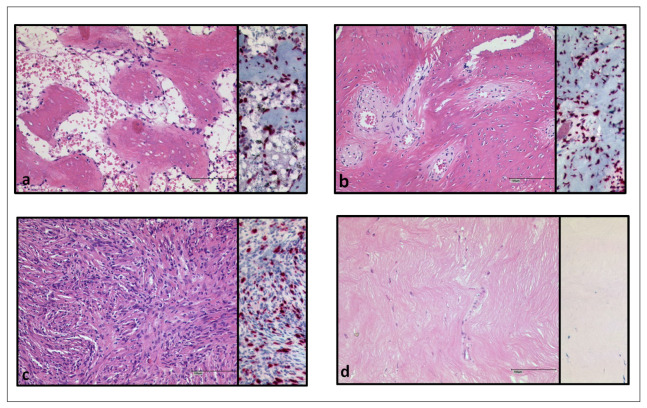
Patterns of matrix formation in H.E. and *H3F3A* G34W staining (**a**) osteoid-like, in trabaeculae (**b**) osteoid-like, roughly organized (**c**) strand-like with spindle shaped cells (**d**) strand-like, fibrosis, very few vital cells, spindle shaped. Scalebar in (**a**–**d**): 100 µm.

**Figure 3 cancers-15-04249-f003:**
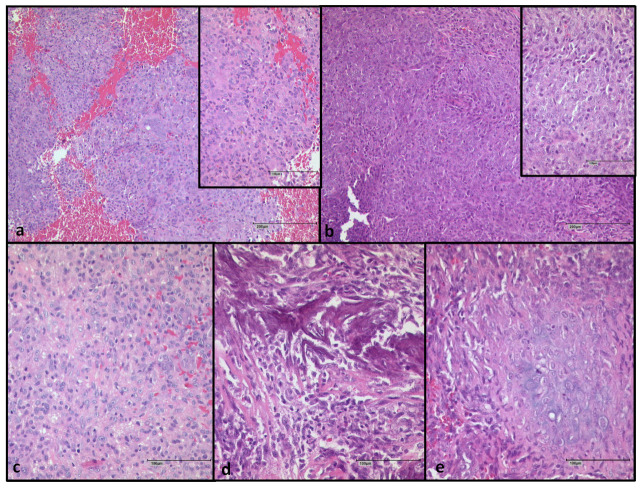
Patient 25 (**a**) Primary giant cell tumor of bone before denosumab therapy (**b**) Osteosarcoma after denosumab therapy (**c**) Osteosarcoma with blastic cells (**d**) with immature osteoid tissue (**e**) chondroblastic cells. Scalebar in (**a**,**b**), image in the back: 200 µm; in (**a**,**b**), small image in front and (**c**–**e**): 100 µm.

**Figure 4 cancers-15-04249-f004:**
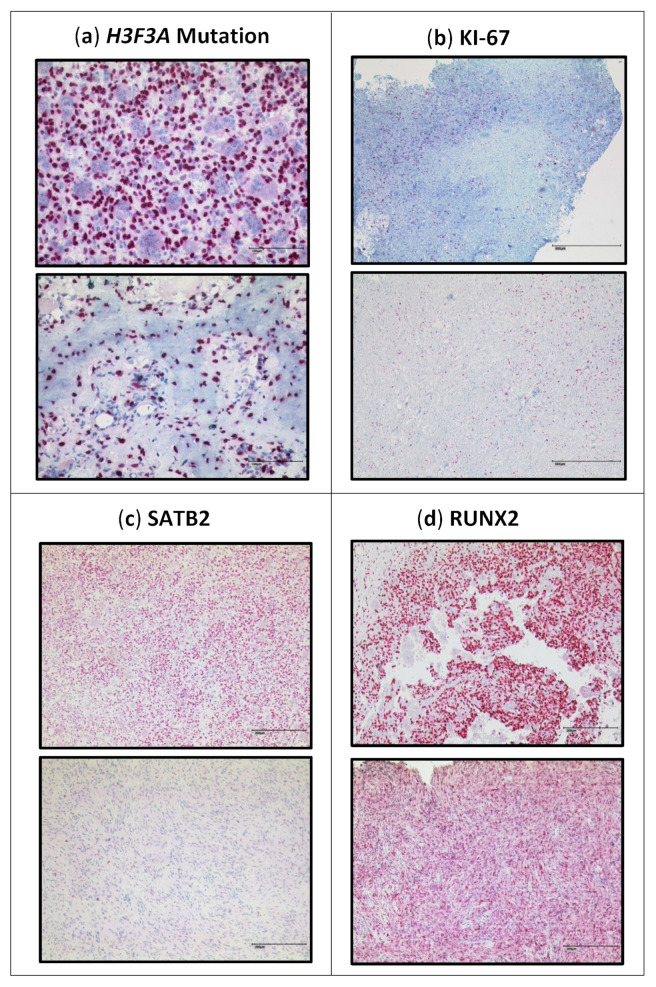
Immunohistochemical staining: upper image before denosumab, lower image after denosumab. (**b**) KI-67 staining, the upper part shows an example of our observation that foci with a high KI-67 index are associated with a higher density of giant cells. Scalebar in (**a**): 100 µm; in (**b**): 500 µm; in (**c**,**d**): 200 µm.

**Figure 5 cancers-15-04249-f005:**
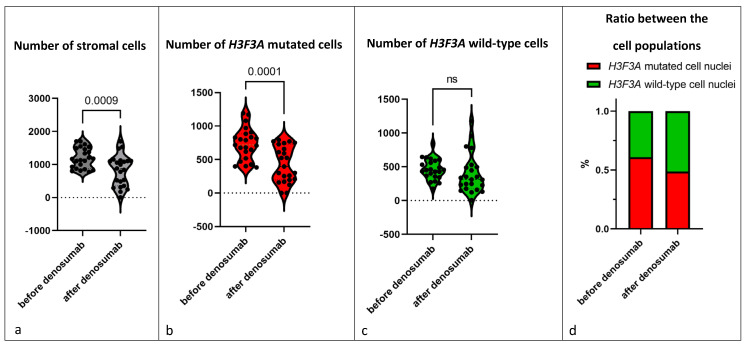
Every individual data point shows the mean of total number of (**a**) stromal cells, (**b**) H3F3A mutated cells and (**c**) H3F3A wild-type cells counted in three representative microscopic fields (200× magnification). Violin diagram is used to show the distribution of the data, small dotted lines show median and quartiles. (**d**) The respective shares of the cell populations in the total stromal cell count. The graphs show all 23 GCTBs treated with denosumab (but without malignant transformation) and compare before and after denosumab therapy.

**Figure 6 cancers-15-04249-f006:**
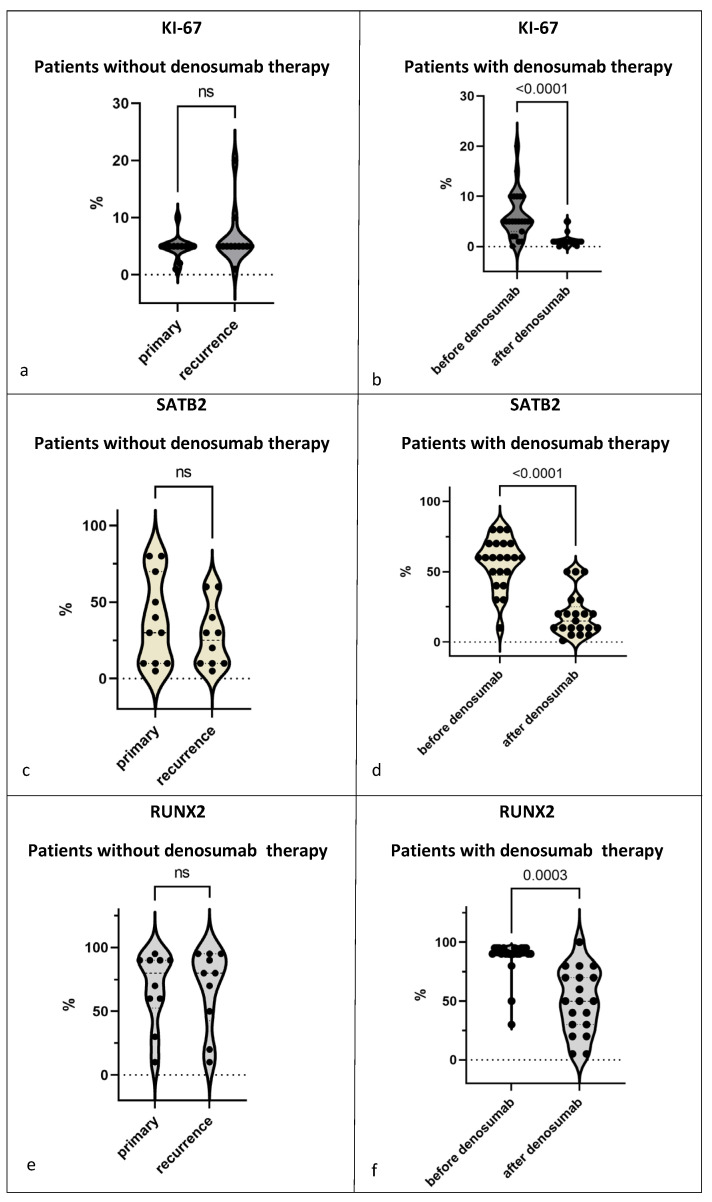
Proportion of positively stained cells in the total number of stromal cells for (**a**,**b**) KI-67: primaries and recurrences *n* = 12, before and after denosumab *n* = 23; (**c**,**d**) SATB2: primaries *n* = 11 and recurrences *n* = 10, before denosumab *n* = 22 and after denosumab *n* = 21 and (**e**,**f**) RUNX2: primaries *n* = 11 and recurrences *n* = 10, before denosumab *n* = 21 and after denosumab *n* = 19. (**a**,**c**,**e**) Comparison of primary and recurrence in the samples of GCTB without denosumab therapy. (**b**,**d**,**f**) Comparison before and after denosumab therapy in the samples of GCTBs treated with denosumab. Violin diagram is used to show the distribution of the data, the individual data points are each the sample of one patient, small dotted lines show median and quartiles.

**Table 1 cancers-15-04249-t001:** Clinical data (sex, tumor localization, duration of denosumab therapy, age at diagnosis) for all 38 patients.

Number	Sex	Tumor Localization	Duration of Denosumab Therapy	Age at Diagnosis
Patient 1	m	lateral condyle of left femur	6 months	40
Patient 2	m	left distal tibia	25 months	37
Patient 3	m	distal radius left	6 months	45
Patient 4	f	spina iliaca anterior	12 months	33
Patient 5	f	proximal femur left	6 months	20
Patient 6	f	lateral femoral condyle left	4 months	28
Patient 7	m	proximal femur right	10 months	52
Patient 8	m	proximal tibia left	20 months	55
Patient 9	m	proximal tibia right	4 months	48
Patient 10	m	left radius	6 months	39
Patient 11	m	distal radius right	8 months	44
Patient 12	f	right patella	9 months	32
Patient 13	f	distal femur right	3 months	26
Patient 14	m	proximal femur left	one single dose	28
Patient 15	f	femoral neck right	12 months	28
Patient 16	f	right os ischii	12 months	20
Patient 17	f	right tibia	4 months	26
Patient 18	m	left hand, phalanx digitus 3	11 months	44
Patient 19	f	proximal tibia right	9 months	13
Patient 20	m	distal femur left	9 months	26
Patient 21	m	proximal tibia left	9 months	31
Patient 22	m	proximal tibia right	2 months	51
Patient 23	f	4th thoracic vertebrae	5 months	32
Patient 24	f	proximal tibia right	12 months	15
Patient 25	m	left femur neck	6 months	28
Patient 26	m	left hand, os metacarpale IV	none	49
Patient 27	m	left hand, phalanx digitus1	none	41
Patient 28	m	right tibia	none	48
Patient 29	f	left tibia	none	63
Patient 30	m	proximal humerus right	none	24
Patient 31	f	proximal tibia left	none	17
Patient 32	f	proximal tibia	none	20
Patient 33	f	proximal tibia right	none	19
Patient 34	m	distal radius left	none	38
Patient 35	f	distal radius right	none	27
Patient 36	m	femur left	none	38
Patient 37	f	femur right	none	27
Patient 38	m	proximal tibia right	none	39

## Data Availability

The data presented in this study are available in this article (and the Appendix A).

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
