# Peer review of "Histomorphometric Analysis of 38 Giant Cell Tumors of Bone after Recurrence as Compared to Changes Following Denosumab Treatment"

_cancers, 2023, doi:10.3390/cancers15174249_

Round 1

Reviewer 1 Report (Previous Reviewer 2)

The authors have addressed all my comments and concerns. The paper is ready for publication. Thank you.

Author Response

First of all, we want to thank again the reviewers for their positive feedback and stimulating thoughts that helped to improve the MS significantly.

REVIEWER 1:

Paper is fine and of interest. The authors have addressed all my comments and concerns. The paper is ready for publication. Thank you.

Answer: We thank the reviewer for reading the MS and his/her positive feedback.

For the complete rebuttal letter please see attachment.

Reviewer 2 Report (New Reviewer)

This paper addresses morphological changes in a quantified way of GCTB following denosumab treatment.

1.     To make the mode of action clear of denosumab and the subsequent morphological changes observed it is important that the readers knows that the multinucleated giant cells are blood born and attracted by the tumour cells. It would be helpful when this would be explained more clearly in the introduction with appropriate references and maybe a cartoon image.

2.     2. More fundamental biology knowledge could be added on the crosstalk between nucleosomes  and telomeres and osteoclastogenesis and epigenetic changes..

3.     There are several recent papers describing the morphological changes after denosumab which could be discussed in order to document the morphological changes and the consistency of it as described in a diverse set of cases.

4.     How does this relate to the changes as for instance seen after treatment of bisphosphonate

5.     malignancy in giant cell tumour is mentioned. Please stick to the WHO terminology and explain (malignancy in and secondary malignant…)

6.     malignancy in giant cells: quite some knowledge is available about the mechanism such as genomic instability and gene expression changes.  Are they substantiated here? The discussion on other molecules known to be involved is lacking.

7.     The simple summary appears not very explanatory and what do the authors mean by markers for surveillance? How would this work?

8.     ORCID ID’s could be added for those authors having one

9.     How do the authors explain the loss of the mutation in case of malignant transformation?

10.  English grammar check is encouraged.

proofreading by a native speaker would be beneficial

Author Response

REVIEWER 2:

We thank the reviewer for the thorough reading and enriching comments. Below we address all of his/her points raised:

  1. To make the mode of action clear of denosumab and the subsequent morphological changes observed it is important that the readers knows that the multinucleated giant cells are blood born and attracted by the tumour cells. It would be helpful when this would be explained more clearly in the introduction with appropriate references and maybe a cartoon image.

Answer: We have added a section regarding this point in the introduction and have added the respective references.

  1. More fundamental biology knowledge could be added on the crosstalk between nucleosomes and telomeres and osteoclastogenesis and epigenetic changes.

Answer: We have added a brief paragraph regarding this argument in the introduction including the corresponding references.

  1. There are several recent papers describing the morphological changes after denosumab which could be discussed in order to document the morphological changes and the consistency of it as described in a diverse set of cases.

Answer: Additionally, to the already mentioned and discussed studies (reference 12, 14, 15, 16, 22-26, 30) we have added three very recent references (reference 27-29) regarding morphological changes after denosumab therapy and placed them in context to our study in the discussion section.

  1. How does this relate to the changes as for instance seen after treatment of bisphosphonate.

Answer: We have added a brief section regarding this argument in the introduction including the corresponding references.

  1. malignancy in giant cell tumour is mentioned. Please stick to the WHO terminology and explain (malignancy in and secondary malignant…)

Answer: We have added a brief paragraph in the introduction explaining malignant giant cell tumor of bone as it is defined by the WHO.

  1. malignancy in giant cells: quite some knowledge is available about the mechanism such as genomic instability and gene expression changes.  Are they substantiated here? The discussion on other molecules known to be involved is lacking.

Answer: We have added a brief paragraph regarding this argument in the introduction including the corresponding references.

  1. The simple summary appears not very explanatory and what do the authors mean by markers for surveillance? How would this work?´

Answer: When we wrote “markers for surveillance” we wanted to emphasize that the markers we analyzed had a clear direction in all benign cases treated with denosumab which was different in our cases with malignant transformation. Therefore, they may have the potential to be an indicator of the course of GCTB. However, more data need to be analysed and added to our limited number of two cases. Therefore, we agree that this complex topic cannot be captured by only a short sentence in the simple summary and abstract and we have deleted it in these sections.

  1. ORCID ID’s could be added for those authors having one

Answer: We have gathered the ORCID numbers of the following authors:

Leinauer, Benedikt: 0000-0002-2516-4787

Pablik, Jessica: 0000-0003-1044-2972

Mogler, Carolin: 0000-0003-3400-7254

Baumhoer, Daniel: 0000-0002-2137-7507

Möller, Peter: 0000-0002-1646-3662

Barth, Thomas FE: 0000-0002-3379-6311

  1. How do the authors explain the loss of the mutation in case of malignant transformation?

Answer: We have added a brief paragraph regarding this argument in the discussion including the corresponding references.

  1. English grammar check is encouraged.

Answer: The MS has been read and corrected by a professional editing service prior to submission.

For the complete rebuttal letter please see attachment.

This manuscript is a resubmission of an earlier submission. The following is a list of the peer review reports and author responses from that submission.

Round 1

Reviewer 1 Report

Paper is fine and of interest.

Results and conclusions are coherent, a longer follow up could be of interest, so I suggest that the paper is considered as a preliminary report. This could open to a following paper about long time results of therapy

Reviewer 2 Report

Specific points and suggestions for improvement of the manuscript are listed below.

General comments:
(1) was analysis done blinded – i.e., recurrent and denosumab slides were coded in a way that the researchers did not know which is which during analysis?

(2) Materials and method section L120-1: “The studied group with recurrences consisted of 7 males and 6 females (median age 34.6); the group with denosumab therapy included 14 males and 11 females (median age 33.6)”. Looking at table 1 and using the ages given three I calculated a median age of 32 to the recurrences group and age of 38 to the denosumab group. This is quite a deviation from the authors’ calculations.

(3) Figure 6c, d, e & f: in each of these violin diagrams I counted the number of data points, and they all fall short from the expected (fewer than n=23 for denosumab group and n=13). Where there any samples excluded from the analysis? This must be clearly stated in the materials and methods section.

(4) Throughout the manuscript: please add years to the publications mentioned in the text. For example: “Girolami et al. conducted…” (L295), should be Girolami et al. (2016).

(5) The authors described 2 types of matrix neoformation in response to Denosumab treatment (results, L146-156). They have discussed these findings again in the discussion (L286-290 and L319-321). However, they never tried to explain these findings and their meaning or explain why one or the other took place in some patients.

(6) See concluding remarks of the summary, abstract and manuscript:

Simple summary (L24-7)

“We detected that profound changes in morphology and the immunohistochemical profile after denosumab therapy are not compatible with changes detected in the sarcomas including expression of RUNX2, SATB2, and KI-67; therefore, these markers may serve as markers for surveillance”.

Abstract (L39-41)

“We conclude that denosumab has a strong impact on the morphology of GCTB. RUNX2 and SATB2 expression differed depending on the benign or malignant course of the tumor under denosumab therapy and may serve as a marker for surveillance”.

Conclusions (L355-7)

“Changes regarding these patterns, such as an increase in KI-67 and increased expression of SATB2 and RUNX2 after denosumab therapy, may point to a malignant transformation associated with treatment and should indicate caution regarding surveillance of these patients”.

Each one is somewhat different and inconsistent with the others. Please edit to deliver the same “take home message” across your text.

Specific comments:

Materials and Methods:

- L113-4: Can you please explain a bit more about your choice of "two-sided paired Student’s t-test"? I’m not sure it is clear to me what was tested here.

Results:

- L123 & 125: “hand”. All other locations are specific bones. Does hand refer to metacarpal or phalanges? I can see the information is given in table 1. Can the author add the specific location to the text?

- L155-159: This section lists the types of matrix neoformation by patients. The authors list 10 + 11 + 2 which sums to 23 patients. Yet the denosumab group had 25 patients. Are the 2 missing patients the ones that developed sarcoma (next paragraph)? This is not clear and should be stated clearly in this section before switching to the next paragraph.

- L173: should be “a complete loss of giant cells”.

- L192: Supplement table S3 is mentioned before supplement table S2 (L205). This makes no sense. Please switch these tables numbering.

- L219: Supplement table S6 is mentioned before supplement table S4 (L227) and S5 (L230). This makes no sense. Please switch these tables numbering.

Discussion:

- L294: “… after 2541 months…”. This will equal more than 211 years of treatment…

Conclusions:

- L353: “… and is lost only after 25 months of treatment with denosumab…”. This was never mentioned or discussed before. New data/results can’t appear in the conclusions for the first time.

Tables:

- Tables S1, S2 and S3 give average values. Please add standard deviation.

Figures:

- Fig 1, 5 and 6: I assume this is a violin diagram? More info needs to be given in the legend on the interpretation of this data visualization. Violin diagrams are used to visualize the distribution of your variable. For example - what is the horizontal dotted line inside the violin diagram? Median, average? All this info must be clearly stated in the figure legend.

- Fig. 2, 3 and 4: please give in the legend the scalebar size. It is impossible to read it from the figure.

- Fig. 5: “Mean of total number…”. I don’t think this is correct. The violin diagram shows the entire population and its distribution, not just the mean value.

- Fig. S1: "Blue: giant cells; cyan: giant cell nuclei". I may be wrong, but I think it should be the opposite (i.e.  "Cyan: giant cells; blue: giant cell nuclei").

References:

- Reference 17 has no year or journal associated with

Reviewer 3 Report

This manuscript described the histology-based analysis of giant cell tumors of bone and evaluated the effect of Denosumab treatment. Collecting 38 tissue samples and conducting a comparative analysis using immunohistochemistry is important to develop a useful prognostic and diagnostic tool. Since H3F3A is already an established marker, the authors looked for other markers by checking the expression of RUNX2, SATB2, and KI-67. Histology is conducted, but little histomorphometric analysis is reported.

Major

·       Simple summary (lines 24-27): The last two sentences do not give a logical summary. It is not logical to conclude as in the text that these markers may serve as markers for surveillance.

·       Selection of markers: Please state the question of this study, and describe the rationale for selecting RUNX2, SATB2, and KI-67.

·       Abstract (lines 39-41): It is stated that Denosumab has a strong impact on the morphology of GCTB. This impact should be described in the abstract and conclusion. The last sentence about the expression of RUNX2, SATB2, and KI-67 is confusing. It is unclear how these markers could serve as a marker.

·       Approach: Histology is conducted but little histomorphometric analysis is included. Please state the reason for choosing histomorphometric analysis and present more about histomorphometric results.

·       Denosumab effect (lines 200-202; lines 304-308): These sentences on the effect of Denosumab are not clear. Please rephrase them.

·       Conclusions: The conclusion is a collection of vague messages, which do not significantly advance our understanding of the effect of Denosumab.

Minor

·       H3F3A mutation (line 87): Please describe the procedure for detecting the mutation.

·       Microscopic field (line 144): Please show the size of the field of the view.

·       2541 months (line 294): Please check the correct duration.

·       5.7 months and 1.5 months [18,22] (lines 328-329): Are the two durations for the two referenced studies?